# New Insights into Xanthophylls and Lipidomic Profile Changes Induced by Glucose Supplementation in the Marine Diatom *Nitzschia laevis*

**DOI:** 10.3390/md20070456

**Published:** 2022-07-14

**Authors:** Xuemei Mao, Xia Wang, Mengdie Ge, Feng Chen, Bin Liu

**Affiliations:** 1Shenzhen Key Laboratory of Marine Microbiome Engineering, Institute for Advanced Study, Shenzhen University, Shenzhen 518060, China; maoxuemei1@163.com (X.M.); xiawang6632@szu.edu.cn (X.W.); supermdge@163.com (M.G.); sfchen@szu.edu.cn (F.C.); 2Institute for Innovative Development of Food Industry, Shenzhen University, Shenzhen 518060, China; 3Institute for Carbon Neutrality, Shenzhen University, Shenzhen 518060, China

**Keywords:** fucoxanthin, eicosapentaenoic acid (EPA), *Nitzschia laevis*, lipidomics, functional foods

## Abstract

*Nitzschia laevis* is a candidate microorganism for bioactive compounds (fucoxanthin and eicosapentaenoic acid (EPA)) production. In this study, the impacts of glucose-induced trophic transition on biomass, photosynthesis, pigments, and lipid profiles were examined. The specific growth rate was increased under glucose addition, achieved at 0.47 day^−1^ (0.26 ± 0.01 day^−1^ for the group without glucose in medium). However, the photosynthetic parameters and pigments including chlorophylls, fucoxanthin, and diatoxanthin were reduced. The net yield of EPA doubled under glucose addition, reaching 20.36 ± 1.22 mg/L in 4 days. In addition, the alteration in detailed lipid molecular species was demonstrated with a focus on EPA-enriched lipids. The effects of 2-deoxyglucose (2DG) indicated that glucose phosphorylation was involved in glucose-induced regulation. These findings provide novel data for guiding the application of this diatom strain in the functional food industries.

## 1. Introduction

Diatoms are promising as natural sources for the production of bioproducts, such as biofuel, pharmaceuticals, and nutraceuticals. For example, fucoxanthin derived from diatoms has antioxidant, anti-inflammatory, anticancer, antidiabetic, antiobesity, and antimetastatic activities [1,2], and EPA could decrease serum triacylglycerol and protect cardiovascular [3,4]. An industrial bottleneck in producing fucoxanthin and EPA is that strains are cultivated autotrophically; hence, the production is space-demanding, and the refinery process is energy-intensive. *Nitzschia laevis* is an emerging chassis with great potential in industrial applications, which accumulates fucoxanthin and eicosapentaenoic acid (EPA), and importantly, it can be cultivated heterotrophically using glucose as an organic carbon source [5].

Xanthophylls, such as fucoxanthin, diadinoxanthin, and diatoxanthin are found in brown seaweeds, diatoms, and dinoflagellates. Fucoxanthin chlorophyll a/c-binding proteins (FCPs) belong to the superfamily of transmembrane light-harvesting complex (LHC) proteins, which direct light energy to the photosystem II (PSII) complexes [6]. Each FCP monomer binds nine chlorophylls (including seven chlorophyll a and two chlorophyll c) and seven fucoxanthins [7]. Chlorophyll c and fucoxanthin enable efficient blue-green light harvesting and energy dissipation in diatoms [7]. The most common carotenogenesis genes in algae and terrestrial plants, e.g., phytoene synthase (PSY) and phytoene desaturase (PDS) have been characterized; however, specific genes for fucoxanthin biosynthesis are yet not clear [8,9]. For the central steps from violaxanthin to fucoxanthin, different pathways were proposed, namely the diadinoxanthin hypothesis and the neoxanthin hypothesis [8]. Fucoxanthin biosynthesis is influenced by environmental factors such as light intensity, salinity, and nutrient availability [10,11,12].

Microalgae as natural sources of *n*-3 long-chain polyunsaturated fatty acids (LC-PUFAs) have the advantages of high content and low contamination, compared with traditional fish oil. The biosynthesis pathway of *n*-3 LC-PUFAs in diatoms has been illustrated in the model system *Phaeodactylum tricornutum* [13]. Acetyl-CoA is converted by the acetyl-CoA carboxylase to malonyl-CoA, which is the intermediate for de novo fatty acids synthesis. The cyclic condensation reactions are catalyzed by fatty acid synthase to extend the acyl group, which is released for further processing by an acyl-ACP thioesterase located on the chloroplast envelope. Then, several elongases and desaturases lead to the formation of EPA. Finally, EPA is incorporated into different lipid classes, including both neutral lipids and polar lipids. The accumulation of EPA and its distribution in lipid species in marine microalgae are responsive to various abiotic conditions such as salinity and nitrogen availability [14,15].

Glucose is a widely consumed organic carbon source for many microorganisms. Glucose enters the microalgal cells in a non-phosphorylated form through glucose transporters and becomes activated by hexokinase before entering central carbon metabolism. Previous studies showed that glucose assimilation has an impact on photosynthesis activity and chloroplast construction in green algae [16]. The docosahexaenoic acid (DHA) and EPA contents in microalgae were also affected by glucose concentration [17,18]. Diatoms grown under phototrophic conditions usually exhibit slow growth and low cell density due to light restrictions [19]. Apart from phototrophic cultivation, *N. laevis* could also be cultivated in mixotrophic or heterotrophic conditions using glucose or other organic carbon sources, e.g., glycerol to reach a higher cell density [5,20]. The trophic flexibility and diversity of high-value compounds make this marine diatom an ideal model for research and industry. However, less attention has been paid to the detailed changes in pigments and lipid profiles responding to glucose addition in *N. laevis*. In the present study, physiological changes were investigated, including biomass, pigments, photosynthesis, and lipidomics, to provide an understanding of the effects of glucose.

## 2. Results and Discussion

### 2.1. Glucose Supplementation Enhances Biomass Accumulation

To examine the impact of glucose on cell growth, we compared the autotrophic culture (AC) with mixotrophic culture (MC). A starting culture grown under autotrophic conditions was used to inoculate a new medium containing 5 g/L glucose, and a paradelle culture without glucose was set up as the control. As shown in Figure 1A, the biomass of MC reached 2.94 ± 0.11 g/L in 4 days, which was 1.30-fold higher than that of AC. The specific growth rate (μ) of the 4-day culture was 0.47 ± 0.01 day^−1^ for the MC group and 0.26 ± 0.01 day^−1^ for the AC group, respectively. The concentration of residual glucose steadily decreased to near zero by the end of day 4, with an overall uptake percentage of 97.4% (Figure 1B). The biomass yield based on the consumption of glucose was 0.51 ± 0.02 g biomass/g glucose in the MC group, which was about 45.7% higher compared with heterotrophic culture in previous studies due to additional carbon fixation from photosynthesis [21].

### 2.2. Cells under Glucose Supplementation Exhibit Reduced Photosynthesis Activity

Glucose supplementation was previously reported to affect photosynthesis and the structure of algal cells [22,23]. In *N. laevis*, the abundance of chlorophyll a (Figure 2A) and chlorophyll c (Figure 2B) was simultaneously decreased in the MC group over the 4 days. As photosynthetic pigments, chlorophyll a plays a role in light harvesting, energy transfer, and light energy conversion, while chlorophyll c in diatom is involved in light harvesting in spite of much smaller amounts than chlorophyll a [24,25]. Meanwhile, to further characterize the photosynthetic activity of mixotrophic and photoautotrophic cells, photosynthesis parameters were examined in detail. The maximum quantum efficiency (Fv/Fm) was quickly decreased in 3 h after the addition of glucose (Figure 2C). The value of F_0_ was raised in 3 h (from 261.7 ± 6.5 in AC to 290.3 ± 9.0 in MC) as the thylakoid membrane reduction state increased due to the supplementation with glucose. The effective quantum yield of PSII photochemistry (Y(II)) and the electron transport rate (ETR) under a given photosynthetically active radiation (PAR = 199) in the MC group were remarkably lower, compared with the control (Figure 2D,E). Figure 2F shows the relative electron transport rate (rETR) changing with PAR, indicating that the limitation in electron transport downstream of PSII was detected in mixotrophic cells. The responses of Fv/Fm, Y(II), and ETR were consistent with the decrease in chlorophylls, reflecting the reduction in photosynthetic capacity after transferring to glucose-feeding cultivation. In addition, as shown in the TEM images (Figure 2G), the thylakoid membranes were less abundant and less compact in MC cells, although the volume of the chloroplasts had no visible defect. Some researchers have reported that glucose has negative impacts on photosynthesis by reducing CO_2_ apparent affinity and chlorophyll contents or repressing the expression of proteins, such as chlorophyll-binding proteins evolved in PSII, the PSII reaction center protein D1, and Rubisco in the Calvin cycle [26,27,28].

### 2.3. Glucose Supplementation Alters Xanthophylls Profiles

Fucoxanthin in *N. laevis* has been identified and quantified by LC–MS [29], while other xanthophylls and their responses to stress have not been reported. The time-resolved changes in fucoxanthin and diadinoxanthin–diatoxanthin (Ddx-Dtx) xanthophylls of *N. laevis* were analyzed in this study (Figure 3). Fucoxanthin content showed a quick response in 3 h after inoculation and was relatively lower in the MC group (Figure 3A). As the fucoxanthin and chlorophylls are bound to FCPs on thylakoid membranes [6,30], the decrease in fucoxanthin under glucose addition was in line with the reduction in chlorophyll contents and photosynthesis parameters. The diadinoxanthin (Ddx) content showed no significant difference between the two groups (Figure 3B), while diatoxanthin (Dtx) content was apparently lower in the MC group (Figure 3C). Hence, the ratio of Ddx/Dtx was significantly higher in the MC group (Figure 3D). The Ddx-Dtx xanthophyll cycle participated in the photoprotection in diatoms by the conversion of Ddx to Dtx under high light, and its reverse reaction occurs in the dark or under weak light [31]. The higher Ddx/Dtx ratio in the MC group is likely a consequence of the photosynthetic apparatus rearrangement, and further work on the epoxidases responsible for the conversion of Dtx and Ddx is worthy of attention. There are two competing hypotheses on fucoxanthin biosynthesis (Figure 3E): One requires the Ddx-Dtx xanthophyll pool to provide the precursor, while the other suggests that Ddx, Dtx, and fucoxanthin are derived from a common precursor (neoxanthin) [32,33]. Thus, the decrease in the total Ddx-Dtx xanthophyll pool suggested the lack of upstream substrates for fucoxanthin biosynthesis.

### 2.4. Mixotrophic Cultivation Improves Fatty Acid Productivity

The total fatty acids (TFAs) and EPA contents in the MC group were significantly higher than those of the AC group (Table 1). TFA and EPA yields were, respectively, increased by 91.3% and 100.6% by glucose addition. From fatty acid (FA) profiling, 13 different fatty acids were identified in *N. laevis*, among which the predominant fatty acids are palmitoleic acid (C16:1), palmitic acid (C16:0), EPA (C20:5), and eicosatetraenoic acid (C20:4), and these 4 FAs constituted up to 74.44% and 76.83% of total fatty acids in the AC and MC groups, respectively. The fatty acids profile in this study showed consistent results with those of previous reports from the same diatom strain that EPA was one of the major fatty acids and positively correlates to glucose concentration [17,34]. The total saturated fatty acid (SFA) content was increased with glucose addition, while that of polyunsaturated fatty acid (PUFA) was decreased mainly due to the reduction in C16:2, C16:3, and C18:4. As a result, the unsaturation level of TFA was lower in the MC group.

A total of 371 lipid molecular species were identified from *N. laevis* using lipidomic analysis according to the saturation status or the length of the carbon chain, belonging to 32 subclasses (Figure 4A), i.e., phosphatidylcholine (PC), triacylglycerol (TG), phosphatidylethanolamine (PE), phosphatidylinositol (PI), phosphatidylglycerol (PG), monogalactosyldiacylglycerol (MGDG), sulfoquinovosyldiacylglycerol (SQDG), digalactosyldiacylglycerol (DGDG), and diacylgyceryl-N-trimethylhomoserine (DGTS). As shown in Figure 4A, PC, TG, MGDG, and SQDG had the most species, contributing to the 52.3% of the total species. Compared with the AC group, differential lipid molecules in the MC group at 48 h were screened using the criteria of fold change ≥ 1.2 or ≤0.83 and *p*-value < 0.05. In summary, there were 38 increased and 27 decreased lipid species (Figure 4B). In addition, 30 most regulated lipid species are shown in Figure 4C, and most of them belonged to saccharolipids and glycerolphospholipids, including 8 MGDGs, 5 lysophosphatidylcholines (LPCs), 4 PCs, 3 PGs, 3 DGDGs, etc. MGDGs, DGDGs, SQDGs, and PGs are the typical lipids of the photosynthetic membrane in higher plants and algae, and MGDG is the main lipid class in the thylakoid and almost undetectable in extraplastidic membrane [35]. Thus, the substantial change in abundance among these species could be a reflection of systematic change in photosynthetic membranes. MGDGs highly enriched in PUFAs are involved in the protection of the thylakoid membrane as a sink of reactive oxygen species [36]. In this study, several PUFA-associated MGDGs (MGDG (16:3/16:3), MGDG (16:1/18:4), MGDG (20:4/20:4), etc.) were decreased, while MGDGs associated with SFAs and monounsaturated fatty acids (MFAs) (MGDG (16:0/18:0), MGDG (16:1/18:0), MGDG (16:1/16:3), etc.) were increased. Among EPA-enriched lipid species, one PG containing EPA (PG (16:0/20:5)) was significantly increased, and one PE and one DGDG containing EPA (PE (16:1/20:5), DGDG (16:3/20:5)) were decreased. In neutral lipids, the contents of TG species containing EPA (TG (16:0/17:1/20:5), TG (16:1/18:1/20:5), TG (18:1/20:4/20:5)), and free EPA remained unchanged, and no diacylglycerol (DG) species containing EPA was detected (Appendix A). Thus, it was speculated that EPAs in polar lipids but not neutral lipids were the main contributor to EPA accumulation under glucose addition. Additionally, previous studies also supported that EPA is mainly present in glycolipids and phospholipids [14].

### 2.5. Glucose Sensing Is Involved in the Regulation of Pigments and Lipids

The glucose transported into cells was phosphorylated by hexokinase to form glucose-6-phosphate, and glucose-6-phosphate was used in various downstream metabolic processes [37]. To investigate whether glucose phosphorylation or the downstream metabolites were involved in the regulation of fucoxanthin and EPA biosynthesis, the effects of a glucose analog, 2-deoxyglucose (2DG), were assessed. Notably, 2DG can be transported into cells and phosphorylated by hexokinase to form 2-deoxyglucose-6-phosphate but cannot be further metabolized [38]. As shown in Figure 5A, the addition of 2DG arrested the growth of the *N. laevis* culture. The analysis of pigments revealed that the abundance of chlorophyll a and fucoxanthin were decreased upon the addition of 2DG (Figure 5B), which indicated that the ability of this diatom to synthesize these pigments or the photosynthetic reaction center complexes was negatively impacted, similar to the impacts of glucose in the mixotrophic cultures. Moreover, the TFA and EPA contents were increased with 2DG addition (Figure 5C). In conclusion, 2DG could induce physiological changes to pigments and fatty acids in *N. laevis* cells similar to those of glucose. Based on the results, a hypothetical model of a glucose-sensing pathway was proposed (Figure 5D). These results suggested that hexokinase-catalyzed glucose phosphorylation was involved in the inhibition of photosynthetic activity and fucoxanthin accumulation, as well as the enhancement of EPA production. Previous studies also reported the central role of hexokinase as a conserved glucose sensor in plants and green algae [39].

## 3. Materials and Methods

### 3.1. Algal Strain and Cultivation Conditions

The marine diatom *N. laevis* (UTEX 2047) was obtained from the Culture Collection of Algae at The University of Texas, Austin, TX, USA, and grown in an LDM medium at 25 °C. Autotrophic and mixotrophic cells were grown in the 250 mL column (3 cm diameter) photoreactor with aeration of 1.5% CO_2_ enriched air and continuous illumination at 40 μmol photons m^−2^·s^−1^. Cells from autotrophic cultures were inoculated into an LDM medium with 5 g/L glucose for mixotrophic transition. For the two-stage strategy in heterotrophic culture, cells were cultivated in a 100 mL LDM medium with 20 g/L glucose in 500 mL flasks in dark, and the glucose was removed in the second stage.

The specific growth rate (μ) was calculated by the equation below:$$\mu =\left(\ln N_{t}-\ln N_{0}\right)/t$$
where *N_t_* is the biomass concentration (g/L) of the culture after *t* days, and *N*_0_ is the initial biomass concentration (g/L).

### 3.2. Biomass and Parameters for Photosynthesis System

The biomass dry weight (DW) was determined by depositing cells on a predried, weighted Whatman GF/C filter paper (0.45 μm pore size) via filtration and dried at 80 °C in a vacuum oven and weighted. Chlorophyll was extracted with 3 mL methanol at 4 °C overnight and centrifuged at 13,000 rpm for 5 min before measurement. By measuring the light absorption at 664 nm and 630 nm with a spectrophotometer, chlorophyll concentration was calculated according to the following equation [40]:Chlorophyll *a* (mg/L) = 13.2654 × A664 − 2.6839 × A630
Chlorophyll *c* (mg/L) = 28.8191 × A630 − 6.0138 × A664

Algal cell suspensions (2 mL) were adapted in the dark for 15 min before measurement of photosynthesis parameters. Photosynthesis parameters, including the maximal quantum efficiency (Fv/Fm), the effective quantum yield of PSII photochemistry (Y(II)), and the electron transport rate (ETR), were determined using a WATER Pulse-Amplitude-Modulation (PAM) equipment (WALZ, Effeltrich, Germany).

### 3.3. Analysis of Xanthophylls

Pigments were extracted from fresh algal cells with methanol. The identification and quantification of carotenoids were performed using an ACQUITY Ultra Performance Liquid Chromatography (UPLC) H-Class coupled in line to a Xevo TQ-XS triple quadrupole mass spectrometer equipped with an ESI probe (Waters, Milford, MA, USA). Samples were injected into an ACQUITY UPLC HSS T3 column (2.1 mm × 100 mm, 1.8 μm, Waters), and the eluents were acetonitrile (A), methanol (B), and demineralized water containing 0.1% (*v*/*v*) formic acid (C). The gradient was initiated from 58% A, 27% B, and 15% C, then transiently to 92% A and 8% C at 4 min, and finally reverted to its initial composition at 8 min, followed by an equilibration phase of 2 min. The flow rate was at 350 μL min^−1^. Nitrogen was used as desolvation gas (800 L/h at 400 °C). The spray capillary voltage was 3000 V for the positive ion mode. Multiple reaction monitoring (MRM) scanning mode was used for mass spectrometry detection. Ions used for the quantification of fucoxanthin, diadinoxanthin, and diatoxanthin were [M + H]^+^ -> 109.1, [M + H]^+^ -> 91.0, and [M + H]^+^ -> 105.1, respectively. The standards employed for the identification and quantification of carotenoids were purchased from Sigma (St. Louis, MO, USA).

### 3.4. Transmission Electron Microscopy (TEM)

Sample preparation for TEM was carried out according to a previous study [41]. Cells were fixed overnight in 2.5% glutaraldehyde buffer (electron microscope purity) at 4 °C. After washing, a mixture of osmium tetroxide and potassium ferricyanide was added and kept at room temperature for 2 h. Then, gradient dehydration was sequentially performed using 30–50–70–90–95–100% alcohol for 10 min per step. After replacing the alcohol with acetone, resin infiltration was performed on a rotary mixer. Epoxy resin with 1.5–2% of the catalyst was injected into an embedding mold, and then the sample was placed in. Polymerization was conducted in an oven at 35 °C for 12 h, followed by 45 °C for 12 h and 60 °C for 24 h. Sample freezing and specimen preparation were conducted according to a previous study [16]. The sample was stained with acetic acid glaze and lead citrate, and subcellular structure was observed using TEM JEM1200EX (JEOL Ltd., Tokyo, Japan) operated at 120 kV.

### 3.5. Lipidomic Analysis

For fatty acids analysis, lyophilized algal cells were transesterified with 1% (*v*/*v*) sulfuric acid in methanol and methylbenzene at 50 °C overnight and analyzed using gas chromatography–mass spectrometry (GC–MS) equipped with a DB-WAX capillary column (30 m × 0.25 mm × 0.25 μm) (Agilent, Santa Clara, CA, USA), as described previously [41]. Heptadecanoic acid (C17:0, Sigma-Aldrich, St. Louis, MO, USA) was used as the internal standard.

The samples for lipidomic analysis were collected via centrifugation and quickly frozen in liquid nitrogen. Total lipids were extracted with prechilled dichloromethane:methanol (3:1, *v*/*v*) after homogenization using a tissue lyser. The supernatant was collected after centrifugation at 25,000 *g* for 15 min at 4 °C. The collected supernatants were vacuum-dried using a Centri Vap benchtop vacuum concentrator and resuspended in isopropanol:acetonitrile:H_2_O (2:1:1, *v*/*v*/*v*). Then, 20 μL of each sample was taken and mixed into a quality control (QC) sample to monitor the reproducibility and stability of the analysis. Lipidomic analysis was performed with a UPLC (Waters, USA) equipped with a Q Exactive high-resolution mass spectrometer (Thermo Fisher Scientific, Waltham, MA, USA) using a CSH C18 column (1.7 μm, 2.1 mm × 100 mm, Waters, USA). The eluents were acetonitrile:H_2_O (6:4, *v*/*v*) containing 10 mM ammonium formate (A) and isopropanol:acetonitrile (9:1, *v*/*v*) containing 10 mM ammonium formate (B). The linear gradient was as follows: 40~43% B over 0~2 min, 43~50% B over 2~2.1 min, 50~54% B over 2.1~7 min, 54~70% B over 7~7.1min, 70~99% B over 7.1~13 min, 99~40% B over 13~13.1 min, constantly held at 99~40% B over 13.1~15 min, and washed with 40% B over 13.1–15min. The flow rate was at 400 μL min^−1^. For positive ion mode, 0.1% (*v*/*v*) formic acid was added to the eluents. MS data were recorded over the range of 200–2000 m/z. The flow rate of sheath gas was 40, and the aux gas flow rate was 10. The spray voltage was 3.8 kV for positive ion mode and 3.2 kV for negative ion mode. The capillary temperature was 320 °C, and the aux gas heater temperature was 350 °C. Data preprocessing including intelligent peak extraction, lipid identification, peak alignment, and statistical analysis was performed with LipidSearch 4.1 software (Thermo Fisher Scientific, USA) and MetaX [42]. The quality deviation of parent ion and fragment ion in the library was set as 5 ppm, and the response threshold was set as the relative response deviation of fragment ion 5.0%. The identification results of lipid molecules using LipidSearch 4.1 are divided into the following four grades: A, B, C, and D. Levels A and B have high accuracy and are, therefore, worth noting. Level A includes lipids that were completely identified with lipid categories and all fatty acid chains. Level B includes lipids with detected category-specific ions and fatty acid fragment ions. Level C includes lipids with only one of class-specific ions or fatty acid fragment ions detected. Level D includes lipids whose lipid structures, such as dehydrated ions could not be recognized.

### 3.6. Statistics Analysis

All sample points were collected from at least three biological replicates. Experimental results are expressed as mean value ± standard deviations (SDs). The statistical significance of the results was validated using a two-tailed *t*-test in the SPSS software (IBM, Armonk, NY, USA).

## 4. Conclusions

Using glucose as an additional carbon source for *N. laevis* cultivation apparently promoted biomass and EPA production. However, glucose induced a decrease in chlorophyll, fucoxanthin, Fv/Fm, Y(II), and ETR, along with the deconstruction of thylakoid structure and the turnover of the membrane lipid, MGDG. The detailed changes in xanthophylls and lipidomic profile of *N. laevis* sheds novel insights into the mechanism regulating fucoxanthin and EPA biosynthesis. Moreover, based on the changes induced by 2DG, a hypothetical model of the glucose-sensing mechanism was summarized. This study elucidated the links between glucose-induced trophic transition and pigments, and PUFA production in *N. laevis* cells, which paves the way for industrial application in the functional food production of this diatom strain.

## Figures and Tables

**Figure 1 marinedrugs-20-00456-f001:**
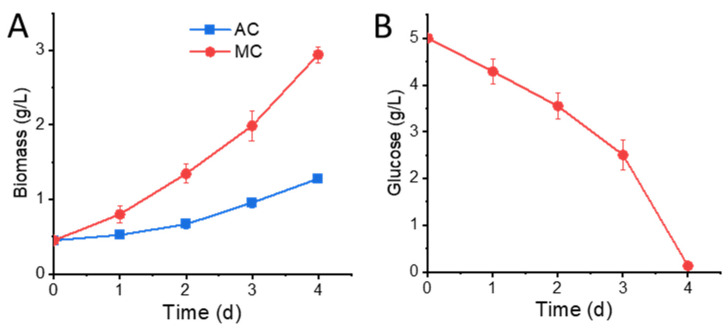
Biomass growth and glucose consumption: (**A**) biomass growth of *Nitzschia laevis* in AC and MC groups; (**B**) residual glucose concentration in MC medium. AC, autotrophic culture; MC, mixotrophic culture.

**Figure 2 marinedrugs-20-00456-f002:**
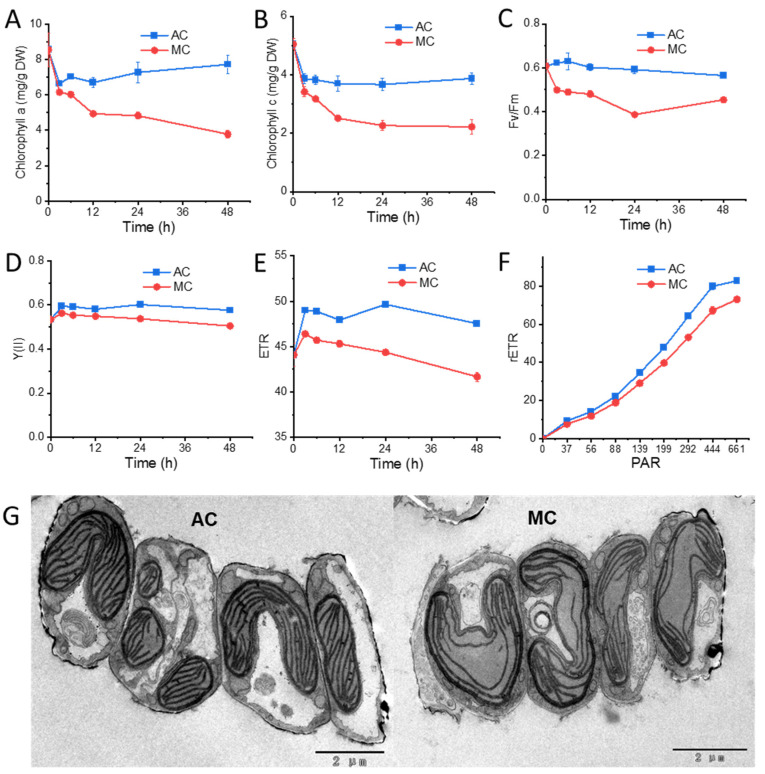
Changes in phototrophic parameters and cell structure in *Nitzschia laevis*: (**A**) chlorophyll a content; (**B**) chlorophyll c content; (**C**) the maximal quantum efficiency (Fv/Fm); (**D**) the effective quantum yield of PSII photochemistry (Y(II)); (**E**) the electron transport rate (ETR) under a given photosynthetically active radiation (PAR = 199); (**F**) the relative electron transport rate (rETR) changing with PAR at 48 h; (**G**) cell structure in TEM images at 48 h. AC, autotrophic culture; MC, mixotrophic culture; DW, dry weight.

**Figure 3 marinedrugs-20-00456-f003:**
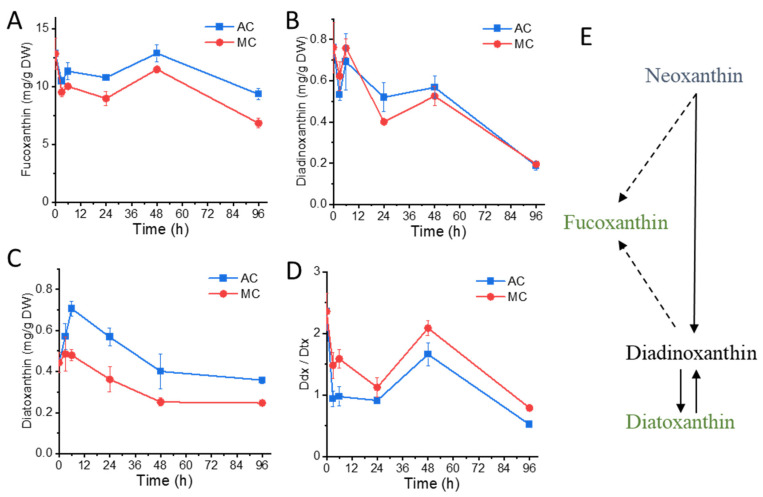
Xanthophylls changes in *Nitzschia laevis*: (**A**) fucoxanthin content; (**B**) diadinoxanthin content; (**C**) diatoxanthin content; (**D**) ratio of diadinoxanthin (Ddx) to diatoxanthin (Dtx); (**E**) hypothetical model of Ddx-Dtx and fucoxanthin changes induced by glucose. The broken line indicates putative pathways for fucoxanthin biosynthesis. The green color indicates the decrease in content. AC, autotrophic culture; MC, mixotrophic culture; DW, dry weight.

**Figure 4 marinedrugs-20-00456-f004:**
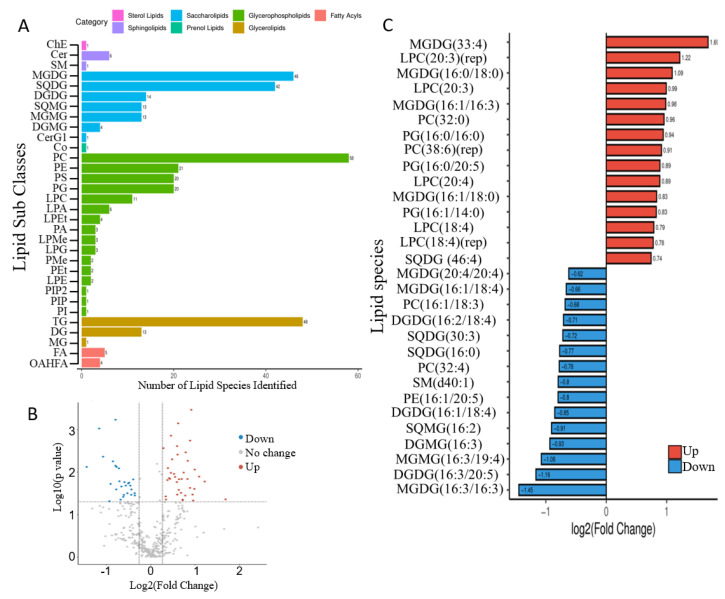
Lipidomic analysis of *Nitzschia laevis* in AC and MC groups at 48 h: (**A**) lipid molecules and subclasses identified via lipidomic analysis; (**B**) the volcano figure shows differential lipid molecules; (**C**) the 30 most regulated lipid species induced by glucose addition.

**Figure 5 marinedrugs-20-00456-f005:**
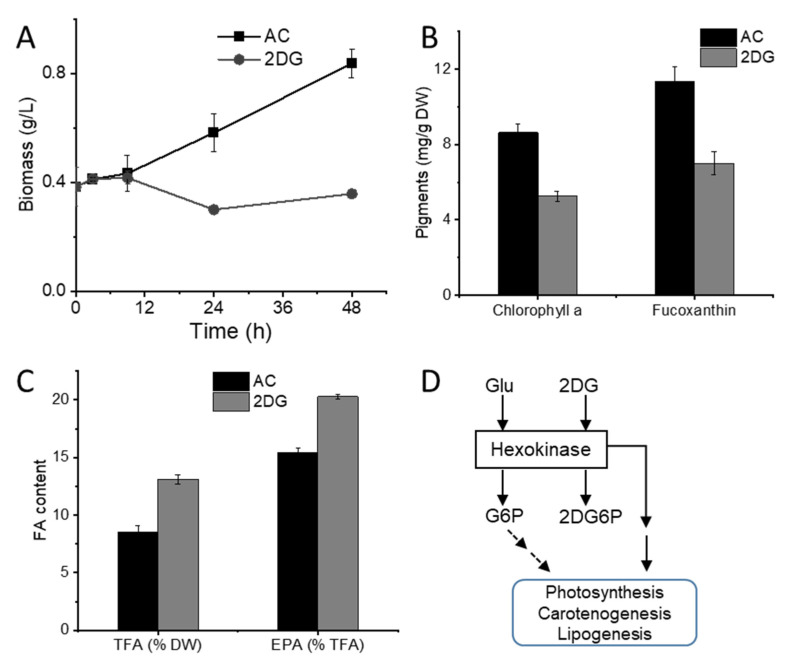
Changes induced by a glucose analog, 2-deoxyglucose (2DG), and the hypothetical model: (**A**) biomass growth of *Nitzschia laevis*; (**B**) chlorophyll and fucoxanthin content; (**C**) TFA and EPA content; (**D**) a hypothetical model of a glucose-sensing pathway. AC, autotrophic culture; DW, dry weight; TFA, total fatty acid; EPA, eicosapentaenoic acid; Glu, glucose; G6P, glucose-6-phosphate; 2DG6P, 2-deoxyglucose-6-phosphate.

**Table 1 marinedrugs-20-00456-t001:** Fatty acids content and profiles in Nitzschia laevis at 4 d under autotrophic (AC) and mixotrophic culture (MC).

Fatty Acids	AC Group ^1^	MC Group ^2^
TFA content (% DW)	8.53 ± 0.54 ^3^	9.42 ± 0.07
TFA yield (mg/L)	65.70 ± 0.82	125.66 ± 3.50
EPA (% DW)	1.31 ± 0.11	1.52 ± 0.06
EPA yield (mg/L)	10.15 ± 0.88	20.36 ± 1.22
C14:0 (% TFA)	3.05 ± 0.02	3.77 ± 0.06
C16:0 (% TFA)	16.78 ± 0.26	19.42 ± 0.34
C16:1 (% TFA)	27.72 ± 0.51	26.74 ± 0.68
C16:2 (% TFA)	9.95 ± 0.19	9 ± 0.52
C16:3 (% TFA)	4.68 ± 0.1	2.38 ± 0.11
C18:0 (% TFA)	0.65 ± 0.04	0.57 ± 0.05
C18:1 (% TFA)	0.84 ± 0.05	1.65 ± 0.09
C18:2 (% TFA)	0.48 ± 0.02	0.94 ± 0.05
C18:3 (% TFA)	2.38 ± 0.04	2.27 ± 0.08
C18:4 (% TFA)	2.18 ± 0.06	1.03 ± 0.04
C20:4 (% TFA)	14.51 ± 0.18	14.47 ± 0.38
C20:5 (% TFA)	15.43 ± 0.37	16.2 ± 0.6
C24:0 (% TFA)	1.33 ± 0.08	1.55 ± 0.01
SFA (% TFA) ^4^	21.82 ± 0.32	25.31 ± 0.38
MFA (% TFA) ^5^	28.56 ± 0.52	28.39 ± 0.74
PUFA (% TFA) ^6^	49.63 ± 0.71	46.3 ± 0.98
∆/mol ^7^	2.15 ± 0.03	2.05 ± 0.03

^1^ AC, autotrophic culture. ^2^ MC, mixotrophic culture. ^3^ values are the means ± standard deviations, *n* = 3. ^4^ SFA, saturated fatty acid. ^5^ MFA, monounsaturated fatty acid. ^6^ PUFA, polyunsaturated fatty acid. ^7^ ∆/mol, degree of unsaturation. ∆/mol = [1.0 (% monoene) + 2.0 (% diene) + 3.0 (% triene) + 4.0 (% tetraene) + 5.0 (% pentaene)]/100.

## Data Availability

Not applicable.

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
