# Peer review of "New Insights into Xanthophylls and Lipidomic Profile Changes Induced by Glucose Supplementation in the Marine Diatom Nitzschia laevis"

_marinedrugs, 2022, doi:10.3390/md20070456_

Round 1

Reviewer 1 Report

The manuscript is overall well organized and written. The experiments were convincingly designed, with proper replication when relevant. Regarding novelty, the effects of mixotrophy onto EPA production by N. laevis have already been extensively described in previous publications. However, the quantification of multiple xantophyll species, the lipidomics dataset and the experiment with 2-deoxyglucose represent previously unreported aspects that give useful insights.

I only have a few specific comments, which are listed hereafter.

Line 32: “Nitzschia laevis is a relatively new marine diatom with great potential in industrial applications…”. The phrasing seems a bit surprising as N. laevis is not per se a “new” diatom species. It would maybe be bette to talk about an “emerging bioindustrial chassis” or something approaching.

Line 60: I think that a sentence about how glucose uptake operates in N. laevis would be useful: glucose enters the cells in a non-phosphorylated form through transporters (are these known?) and becomes activated by hexokinase before entering central carbon metabolism.

Line 99: “The maximum quantum efficiency (Fv/Fm) was quickly decreased in 3 hours after the addition of glucose (Fig. 2C).” Did the authors look at the values of Fm and F0 separately? F0 could rise if the thylakoid membrane reduction state increases, which could happen due to the supplementation with glucose.

Line 101: “the electron transport rate (ETR) under a given photosynthetically active radiation (PAR)” Which PAR exactly?

Line 103: “The higher Ddx/Dtx ratio in the MC group could be the result of the cell-shading effect due to increased cell density.” This is unlikely because after 6h the ratio already strongly differed between AC and MC whereas no major differences in growth (self-shading) occurred. I would to think this is a logic consequence of the photosynthetic apparatus rearrangement, maybe due to a downregulation of the epoxidases responsible for dtx conversion into ddx.

Figure 4: In the reviewer version, the definition of the figure is pretty low which makes the lipid species names (panels A and C) somehow blurry.

Panel A: I originally thought this was an absolute lipid quantification because of the X axis label “Lipid Molecules Number”. Could you change this label to something like “Number of Lipid Species Identified”?

Author Response

The manuscript is overall well organized and written. The experiments were convincingly designed, with proper replication when relevant. Regarding novelty, the effects of mixotrophy onto EPA production by Nlaevis have already been extensively described in previous publications. However, the quantification of multiple xantophyll species, the lipidomics dataset and the experiment with 2-deoxyglucose represent previously unreported aspects that give useful insights.

I only have a few specific comments, which are listed hereafter.

Line 32: “Nitzschia laevis is a relatively new marine diatom with great potential in industrial applications…”. The phrasing seems a bit surprising as Nlaevis is not per se a “new” diatom species. It would maybe be bette to talk about an “emerging bioindustrial chassis” or something approaching.

Response:

Thanks for your suggestion. We have revised this sentence as ‘Nitzschia laevis is an emerging chassis with great potential in industrial applications. (Line 32, in red)

Line 60: I think that a sentence about how glucose uptake operates in Nlaevis would be useful: glucose enters the cells in a non-phosphorylated form through transporters (are these known?) and becomes activated by hexokinase before entering central carbon metabolism.

Response:

The sentence has been added according to the reviewer’s suggestion. ‘Glucose enters the microalgal cells in a non-phosphorylated form through glucose transporters and becomes activated by hexokinase before entering central carbon metabolism.’ Line 61, in red.

Line 99: “The maximum quantum efficiency (Fv/Fm) was quickly decreased in 3 hours after the addition of glucose (Fig. 2C).” Did the authors look at the values of Fm and F0 separately? F0 could rise if the thylakoid membrane reduction state increases, which could happen due to the supplementation with glucose.

Response:

Yes, we investigated the value of F0. It was raised in 3 hours with glucose supplementation. On the contrary, the value of Fm decreased. Considering the reviewer’s comments, we also added the description in the text. ‘The value of F0 was raised in 3 hours (from 261.7± 6.5 in AC to 290.3 ± 9.0 in MC) as the thylakoid membrane reduction state increased due to the supplementation with glucose.’ Line 102, in red.

Line 101: “the electron transport rate (ETR) under a given photosynthetically active radiation (PAR)” Which PAR exactly?

Response:

We have added the PAR value in the text. (Line 106, in red) The PAR was 199 which was a fixed PAR intensity for the light induction function of the PAM system.

Line 103: “The higher Ddx/Dtx ratio in the MC group could be the result of the cell-shading effect due to increased cell density.” This is unlikely because after 6h the ratio already strongly differed between AC and MC whereas no major differences in growth (self-shading) occurred. I would to think this is a logic consequence of the photosynthetic apparatus rearrangement, maybe due to a downregulation of the epoxidases responsible for dtx conversion into ddx.

Response:

Thank for your valuable suggestion. Here, we have revised the discussion according to your comments. ‘The higher Ddx/Dtx ratio in the MC group is likely a consequence of the photosynthetic apparatus rearrangement, and further work on the epoxidases responsible for the conversion of Dtx and Ddx is worth attention.’ (Line 138, in red)

Figure 4: In the reviewer version, the definition of the figure is pretty low which makes the lipid species names (panels A and C) somehow blurry.

Response:

Considering the reviewer’s suggestion, Figure 4 has been updated with a higher definition.

Panel A: I originally thought this was an absolute lipid quantification because of the X axis label “Lipid Molecules Number”. Could you change this label to something like “Number of Lipid Species Identified”?

Response:

The label of X axis in Figure 4A has been changed to “Number of Lipid Species Identified”.

Thank you very much for your comments and suggestions.

Reviewer 2 Report

This study reports the growth of and EPA production by the diatom Nitzschia laevis under mixotrophic culture (MC) with glucose as compared with autotrophic cultivation (AC). Data repoted in the study suggest improvement of EPA production under MC. The experimental set-up is well conducted and data conveniently presented in general terms. The study is of interest for readers of this journal. Nonetheless, the issues that follow should be properly addressed before the manuscript is definitely accepted:

1) Abstract, lines17-20: rewrite avoiding ambigueties (not the underlying mechanism is studied but the implication of one enzyme exclusively, for instance).

2) Figure 2E: please indicate the PAR to which the ETR was measured.

3) Figure 2F: why were the measurements done at 48 h?

4) Figure 5: comparison with MC (using glucose) instead of AC is missing

5) Materials and methods, section 3.2: please indicate if the measurements of chlorophyll fluorescence were done over all culture or an aliquote was withdrawn.

6) Materials and methods, section 3.3: please indicate the adducts that were detected for every xanthophyll

7) Materials and methods, section 3.5: please avoid using the term "lipidomic" throughout the text. If you want to refer to the discipline, "lipidomics" should be used. I you intend to refer to the compound class, the terms "lipid" or "lipidic" should be used.

8) Materials and methods, section 3.5: explain better how the lipids were measured: were all of them measured in one run?, was there separation in the chromatogram between the different classes? Please afford a figure with chromatograms for the AC and MC.

9) Materials and methods, section 3.5: How were the regioisomers identified? That is, which criterium was used to assign the acyl chains to the different sn- positions of the glycerol, in particular for triacylglycerols. Were standards used for every lipid class?

10) Materials and methods, section 3.6: was two-tailed t-test? Please indicate the software used for the statistical analysis.

11) Supplementary material (EXCEL file): please indicate what "NA" means in the column P ("MainIon1")

Author Response

This study reports the growth of and EPA production by the diatom Nitzschia laevis under mixotrophic culture (MC) with glucose as compared with autotrophic cultivation (AC). Data repoted in the study suggest improvement of EPA production under MC. The experimental set-up is well conducted and data conveniently presented in general terms. The study is of interest for readers of this journal. Nonetheless, the issues that follow should be properly addressed before the manuscript is definitely accepted:

1) Abstract, lines17-20: rewrite avoiding ambigueties (not the underlying mechanism is studied but the implication of one enzyme exclusively, for instance).

Response:

According to the reviewer’s suggestion, we have revised the ambiguous sentence and deleted the words referring to ‘the underlying mechanism’. Line 18.

2) Figure 2E: please indicate the PAR to which the ETR was measured.

Response:

The PAR value has been added for Figure 2E in the figure caption (line122, in red).

3) Figure 2F: why were the measurements done at 48 h?

Response:

The results on chlorophyll, the effective quantum yield of PSII photochemistry (Y(II)) and the electron transport rate (ETR) indicated that the state of the photosynthetic system was obviously changed with glucose supplement, reaching the lowest value at 48 h. And Figure 2E has shown the change of ETR over time, which is measured under one PAR intensity. For more details, we measured rETR changes under a series PAR intensity.

4) Figure 5: comparison with MC (using glucose) instead of AC is missing

Response:

Here, we compared the culture adding 2-deoxyglucose (2DG) with the AC without 2DG to investigate the roles of glucose phosphorylation. Since the relevant comparison for MC and AC has been discussed thoroughly in previous sections, we didn’t add MC in Figure 5 again to avoid repetition.

5) Materials and methods, section 3.2: please indicate if the measurements of chlorophyll fluorescence were done over all culture or an aliquote was withdrawn.

Response:

Considering the reviewer’s comment, we have added the volume (2 mL) of the aliquot used for the measurements of chlorophyll fluorescence in this section. (Line 254, in red)

6) Materials and methods, section 3.3: please indicate the adducts that were detected for every xanthophyll

Response:

Thanks for your valuable suggestion. We have added that ‘Ions used for the quantification of fucoxanthin, diadinoxanthin and diatoxanthin were [M+H]+ -> 109.1, [M+H]+ -> 91.0 and [M+H]+ -> 105.1, respectively’. Line 271, in red.

7) Materials and methods, section 3.5: please avoid using the term "lipidomic" throughout the text. If you want to refer to the discipline, "lipidomics" should be used. I you intend to refer to the compound class, the terms "lipid" or "lipidic" should be used.

Response:

We have revised the term ‘lipidomic’ in the text according to your suggestion. Line 286, 172, 202, 203, 300, 332.

8) Materials and methods, section 3.5: explain better how the lipids were measured: were all of them measured in one run?, was there separation in the chromatogram between the different classes? Please afford a figure with chromatograms for the AC and MC.

Response:

Methods have been revised accordingly (Line 304-307, 312, in red). Lipids were measured in two runs: one in positive ion mode and another in negative ion mode. The chromatographic conditions have been explained in section 3.5 (Line 304-307). The chromatogram provided the primary separation for lipids, and then, LC- MS/MS data processing including intelligent peak extraction, lipid identification, and peak alignment carried out by LipidSearch 4.1 software (Thermo Fisher Scientific, USA) (Line 312). According to the reviewer’s suggestion, we added the chromatograms of base peak ion for the AC and MC in Supplementary file 2.

9) Materials and methods, section 3.5: How were the regioisomers identified? That is, which criterium was used to assign the acyl chains to the different sn- positions of the glycerol, in particular for triacylglycerols. Were standards used for every lipid class?

Response:

The methods in the text have been revised accordingly and more details have been added (Line 313-321, in red). The raw data generated by LC-MS/MS detection was imported into LipidSearch v.4.1(Thermo Fisher Scientific, USA) for lipid molecular identification. The database of this software included information about 1.7 million lipid ions (including many regioisomers) and their predicted fragment ions. For the standards of lipid identification, the quality deviation of parent ion and fragment ion in the library was set as 5 ppm, and the response threshold was set as the relative response deviation of fragment ion 5.0%. All identification results of lipid molecules by LipidSearch 4.1 are divided into the following four grades (A, B, C and D), in case the identification results of some regioisomers may not be convinced. LevelA and B have high accuracy and can be paid attention to. LevelA: Lipid categories and all fatty acid chains can be completely identified. LevelB: Category-specific ions and fatty acid fragment ions can be detected. Level C: Class-specific ions or fatty acid fragment ions, only one of which can be detected. Level D: Lipid structures, such as dehydrated ions, cannot be recognized.

10) Materials and methods, section 3.6: was two-tailed t-test? Please indicate the software used for the statistical analysis.

Response:

The methods have been revised. ‘The statistical significance of the results was validated using a two-tailed t-test in the software SPSS.’ Line 326, in red.

11) Supplementary material (EXCEL file): please indicate what "NA" means in the column P ("MainIon1")

Response:

The explanation has been added in supplementary material 1. Column P ("MainIon1") indicates fragment ions used for lipid quantification. Thus, the ‘NA’ in this column means no fragment ion (using parent ions instead) was used for the quantification.

Thank you very much for your comments and suggestions.